# CK2α/CSNK2A1 Induces Resistance to Doxorubicin through SIRT6-Mediated Activation of the DNA Damage Repair Pathway

**DOI:** 10.3390/cells10071770

**Published:** 2021-07-13

**Authors:** Usama Khamis Hussein, Asmaa Gamal Ahmed, Yiping Song, Kyoung Min Kim, Young Jae Moon, Ae-Ri Ahn, Ho Sung Park, Su Jin Ahn, See-Hyoung Park, Jung Ryul Kim, Kyu Yun Jang

**Affiliations:** 1Department of Pathology, Jeonbuk National University Medical School, Jeonju 54896, Korea; usamahussein@jbnu.ac.kr (U.K.H.); asmaascience3@gmail.com (A.G.A.); kmkim@jbnu.ac.kr (K.M.K.); xoxoxyool@naver.com (A.-R.A.); hspark@jbnu.ac.kr (H.S.P.); 2Research Institute of Clinical Medicine of Jeonbuk National University-Biomedical Research Institute of Jeonbuk National University Hospital, Jeonju 54907, Korea; yjmoonos@jbnu.ac.kr; 3Faculty of Science, Beni-Suef University, Beni-Suef 62511, Egypt; 4Faculty of Postgraduate Studies for Advanced Sciences, Beni-Suef University, Beni-Suef 62511, Egypt; 5Department of Orthopedic Surgery, Jeonbuk National University Medical School, Jeonju 54896, Korea; soybeanhan@gmail.com; 6Department of Biochemistry and Molecular Biology, Jeonbuk National University Medical School, Jeonju 54896, Korea; asujin2443@gmail.com; 7Department of Bio and Chemical Engineering, Hongik University, Sejong 30016, Korea

**Keywords:** osteosarcoma, CSNK2A1, SIRT6, doxorubicin, DNA damage, prognosis

## Abstract

CK2α/CSNK2A1 is involved in cancer progression by phosphorylating various signaling molecules. Considering the role of CSNK2A1 in cancer progression and the phosphorylation of SIRT6 and the role of SIRT6 in chemoresistance through the DNA damage repair pathway, CSNK2A1 and SIRT6 might be involved in resistance to conventional anti-cancer therapies. We evaluated the expression of CSNK2A1 and phosphorylated SIRT6 in the 37 osteosarcoma patients and investigated the effects of CSNK2A1 and the phosphorylation of SIRT6 on Ser338 on resistance to the anti-cancer effects of doxorubicin. Higher expression of CSNK2A1 and phosphorylated SIRT6 was associated with shorter survival in osteosarcoma patients. U2OS and KHOS/NP osteosarcoma cells with induced overexpression of CSNK2A1 were resistant to the cytotoxic effects of doxorubicin, and the knock-down of CSNK2A1 potentiated the cytotoxic effects of doxorubicin. CSNK2A1 overexpression-mediated resistance to doxorubicin was associated with SIRT6 phosphorylation and the induction of the DNA damage repair pathway molecules. CSNK2A1- and SIRT6-mediated resistance to doxorubicin in vivo was attenuated via mutation of SIRT6 at the Ser338 phosphorylation site. Emodin, a CSNK2A1 inhibitor, potentiated the cytotoxic effects of doxorubicin in osteosarcoma cells. This study suggests that blocking the CSNK2A1-SIRT6-DNA damage repair pathway might be a new therapeutic stratagem for osteosarcomas.

## 1. Introduction

Protein kinase CK2 (designated CSNK2) is a highly conserved serine/threonine kinase with diverse roles in the cell and is involved in the regulation of more than 300 molecules [1]. CSNK2 is especially involved in tumor biology through regulation of cellular proliferation, cell-cycle progression, apoptosis, metabolism, and invasiveness [1,2,3]. CSNK2 α1 (CSNK2A1, CK2α) expression is also elevated in cancer tissue of the gastrointestinal tract [4,5], head and neck [6], kidney [7,8], and prostate [9]. Furthermore, higher expression of CSNK2A1 is associated with shorter survival in patients with breast cancer [10], clear cell renal cell carcinoma [8], and gastric cancer [5]. CSNK2A1 is involved in cancer progression by regulating various cancer-promoting signaling pathways, such as the MYC, PI3K-Akt, NFκB, and Wnt/β-catenin pathways [3,10,11,12,13,14]. These biological roles of CSNK2A1 in cancer are closely related to its kinase activity on the targets involved in tumorigenesis [5,10]. CSNK2A1 serves cancer progression by inducing the phosphorylation of various molecules, including SIRT1, SIRT6, and CCAR2 [5,10,15]. Among these, a study on the phosphorylation of SIRT6 has focused on its important roles in cancer progression [10]. 

SIRT6 is a member of the sirtuin family and has diverse roles in normal physiology and cancer biology, including aging, cell metabolism, proliferation, the invasiveness of cells, and DNA damage repair [10,16,17,18]. Among these, the role of SIRT6 in DNA damage repair has suggested a role for SIRT6 as a tumor suppressor [18]. However, controversially, poor prognosis of cancer patients having a higher expression of SIRT6 has been reported in various human cancers, such as breast cancer [10], gastric cancer [19], lung cancer [20], osteosarcoma [21], and ovarian cancer [22]. The role of SIRT6 in DNA damage repair enables cancer cells to be refractory genotoxic anti-cancer therapeutics [21]. In osteosarcoma cells, the inhibition of SIRT6 sensitized cells to the cytotoxic effect of doxorubicin [21]. Therefore, therapies targeting SIRT6 or the downstream DNA damage repair pathway have been suggested for cancers highly expressing SIRT6. However, when considering the controversial reports on the effects of SIRT6 in cancer cells [10,21,23,24], the elevated expression of SIRT6 might not be sufficient to predict SIRT6 activity in human cancers. The effects of SIRT6 on cancer biology are regulated by its phosphorylation status, especially on Ser338 [10]. In breast cancer cells, CSNK2A1 phosphorylates SIRT6 on Ser338, and this process stimulates cell proliferation through the regulation of β-catenin and NFκB [10]. Therefore, the relationship between CSNK2A1 and SIRT6 phosphorylation on Ser338 might be important in cancer therapy with regards to overcoming resistance to anti-cancer therapeutics. 

Osteosarcoma is the most common primary cancer of the bone [25]. However, despite advancements in the understanding of the molecular mechanisms of cancer and the development of specific anti-cancer targeted therapies, therapeutic modalities applicable to osteosarcoma are limited [26]. Therefore, trials to find specific therapeutic applications focused on osteosarcoma are needed. Recently, therapeutic applications targeting the DNA damage repair pathway by using a PARP inhibitor [27,28] and by inhibiting SIRT6 [21] have been reported in osteosarcoma. In addition, the efficacy of the inhibition of CSNK2 has been evaluated in various human cancers [29,30]. Therefore, based on the roles and molecular relationships between CSNK2A1-SIRT6-DNA damage repair pathways in human cancers [10,17,21], this molecular relationship might be a potential therapeutic target for osteosarcoma. However, reports on the role of CSNK2A1-SIRT6 pathway in human sarcoma, including osteosarcoma, are limited. Therefore, this study aimed to investigate the roles and mechanisms of CSNK2A1-SIRT6-DNA damage repair pathways in the treatment of osteosarcomas, especially those resistant to anti-cancer therapy.

## 2. Materials and Methods

### 2.1. Osteosarcoma Specimens and Immunohistochemical Evaluation of CSNK2A1

Thirty-seven osteosarcomas of the bone treated between 1999 and 2011 at Jeonbuk National University Hospital were included in this study. The cases included in this study were evaluated for clinicopathological factors by reviewing medical records and histologic slides. A review of the case was based on the latest WHO classification [25] and the latest AJCC cancer staging system [31]. Twenty-six patients received adjuvant chemotherapy (doxorubicin, high-dose methotrexate, and cisplatin). The clinicopathological factors evaluated in this study are listed in Table 1. The expression of CSNK2A1 in human osteosarcoma tissue samples was assessed via immunohistochemical staining of tissue microarray (TMA) sections. TMA blocks contained two 3 mm tissue cores per case. TMA sections were microwaved using an antigen retrieval procedure with pH 6.0 antigen retrieval solution (Cat. #; S1699, DAKO, Glostrup, Denmark). Tissue sections were incubated with anti-CSNK2A1 (1:100, Cat. #; 2656, Cell Signaling Technology, Beverly, MA, USA) and phosphorylated SIRT6 (Ser338) (pSIRT6, 1:100, Cat. #; 9B06v17, Cell Signaling Technology, Beverly, MA, USA) antibodies and developed with the DAKO Envision system (Cat. #; K4001, DAKO, Carpinteria, CA, USA). Immunohistochemical staining scores were obtained by adding the intensity score (no expression: score 0, weak expression: score 1, moderate expression: score 2, strong expression: score 3) and the area score (no staining cells: score 0, 1%: score 1, 2~10%: score 2, 11~33%: score 3, 34~66%: score 4, and 67~100%: score 5) [21,32,33,34]. Thereafter, the scores obtained from each TMA section from two TMA cores in each case were added and used in the analysis. The final scores ranged from zero to sixteen. This study was performed after obtaining IRB approval (Jeonbuk National University Hospital; approval number, CUH 2020-10-011, date of approval: 16 October 2020).

### 2.2. Osteosarcoma Cells, Chemicals, and Transfection 

This study used two human osteosarcoma cell lines. U2OS cells were purchased from the Korean Cell Line Bank (KCLB, Seoul, Korea), and KHOS/NP cells were kindly provided by Chang-Bae Kong (Department of Orthopedic Surgery, Korea Institute of Radiological and Medical Science). U2OS and KHOS/NP cells were grown in Dulbecco’s modified Eagle’s medium (Cat. #; LM001-05, Gibco BRL, Gaithersburg, MD, USA) under humidified conditions of 5% CO_2_ and 37 °C. The culture media were supplemented with 10% Fetal Bovine Serum (Cat. #; 09320001, Gibco BRL) and 1% penicillin/streptomycin (Cat. #; 15240-062, Gibco BRL). Cells were routinely assessed for mycoplasma contamination with the Mycotest kit (Invitrogen, Carlsbad, CA, USA). This study used doxorubicin (D1515, Cat. #; 25316-40.9, Sigma, St. Louis, MO, USA) and emodin, a CSNK2A1 inhibitor (Cat. #; 043K35051V, Sigma, St. Louis, MO, USA). The vector for CSNK2A1-specific shRNA was purchased from GenePharma Co. (Cat. #; A08500, GenePharma, Shanghai, China). The CSNK2A1 duplex has the forward and reverse sequences 5′-CACCGGGTGAAACACTTCAGAAGCATTCAAGAGATGCTTCTGAAGTGTTTCACCCTTTTTTG-3′ and 5′-GATCCAAAAAAGGGTGAAACACTTCAGAAGCATCTCTTGAATGCTTCTGAAGTGTTTCACCC -3′, respectively. The plasmid for HA-tagged wild-type (WT) *CSNK2A1* (pRc/CMV_CSNK2A1-HA) was kindly provided by Laszlo Gyenis (Department of Biochemistry, University of Western Ontario, Canada). The plasmids for WT-*SIRT6* (pFLAG2_SIRT6) and mutant construct for the Ser338 phosphorylation site of SIRT6 (pFLAG2_SIRT6_S338A) were synthesized by Cosmo Genetech Co. Ltd. (Seoul, Korea). A pFLAG-CMV-2 vector was used as a control plasmid. Lipofectamine 2000 (Cat. #; 11668-019, Invitrogen, Carlsbad, CA, USA) was used for the transfection of cells.

### 2.3. Cell Proliferation Assay and Colony-Forming Assay

The proliferation activity of cells was evaluated by using the Cell Counting Kit-8 assay (CCK8, Cat. #; PK648, Dojindo, Kumamoto, Japan) and a colony-forming assay. The CCK8 assay was performed by seeding 3 × 10^3^ U2OS and 3 × 10^3^ KHOS/NP cells in 96-well plates for 24, 48, and 72 h. Two hours after adding CCK8, the absorbance was measured at 560 nm in a microtiter plate reader (Bio-Rad, Richmond, CA, USA). The colony-forming assay was performed by seeding 1 × 10^3^ U2OS and 1 × 10^3^ KHOS/NP cells in 12-well or 6-well culture plates. After the indicated times, the plates with colonies were fixed in methanol and stained with 0.1% crystal violet (Cat. #; 548-62-9, Sigma, St. Louis, MO, USA), then counted using Clono-Counter software available as supplementary electronic material of a published manuscript [35].

### 2.4. Western Blot Analysis

To obtain the total protein, we washed cells with phosphate-buffered saline twice and lysed them with ice-cold PRO-PREP Protein Extraction Solution (Cat. #; 17081, iNtRON Biotechnology Inc., Seong-nam, Korea) supplemented with 1x phosphatase inhibitor cocktails 2, 3 (Cat. #; 108M4000V, Sigma, St. Louis, MO, USA). The normalized protein was mixed with 4× SDS-PAGE loading buffer and electrophoresed. The electrophoresed proteins were electrotransferred to a polyvinylidene difluoride membrane and blocked. The membranes were incubated with the following primary antibodies: CSNK2A1 (Cat. #; 2656, Cell Signaling Technology, Beverly, MA, USA), SIRT6 (Cat. #; 2486, Cell Signaling Technology, Beverly, MA, USA), phosphorylated SIRT6 (Ser338) (pSIRT6, Cat. #; 9B06v17, Cell Signaling Technology, Beverly, MA, USA), PARP1 (Cat. #; sc-8007, Santa Cruz Biotechnology, Santa Cruz, CA, USA), cleaved PARP1 (Cat. #; 5625, Cell Signaling Technology, Beverly, MA, USA), cleaved caspase 3 (Cat. #; 9664, Cell Signaling Technology, Beverly, MA, USA), BCL2 (Cat. #; 3498, Cell Signaling Technology, Beverly, MA, USA), BAX (Cat. #; 2774, Cell Signaling Technology, Beverly, MA, USA), phosphorylated ATM (Ser1981) (pATM, Cat. #; sc-47739, Santa Cruz Biotechnology), phosphorylated Chk2 (pChk2, Cat. #; 2197, Cell Signaling Technology, Beverly, MA, USA), phosphorylated H2AX (γH2AX, Cat. #; 9718, Cell Signaling Technology, Beverly, MA, USA), and GAPDH (Cat. #; 5174, Cell Signaling Technology, Beverly, MA, USA). The membranes were incubated with secondary antibodies and developed with an ECL detection system (Cat. #; WBKL50500, Amersham Biosciences, Buckinghamshire, UK). The developed membranes were imaged by using a luminescent image analyzer (LAS-3000, Fuji Film, Tokyo, Japan). The bands of the western blots were quantified by using ImageJ software (https://imagej.nih.gov/ij, accessed on 3 July 2021).

### 2.5. Flow Cytometry Analysis for Apoptosis

The apoptosis of cells was measured using an apoptosis detection kit (Cat. #; 556547, Invitrogen, Carlsbad, CA, USA) via flow cytometry analysis based on staining for FITC-conjugated annexin V and propidium iodide. The cells were suspended in 100 μL binding buffer at a concentration of 1 × 10^6^ cell/mL. The suspended cells were incubated with 5 μL of annexin V-FITC and 5 μL of propidium iodide for 15 min at 37 °C in the dark. Thereafter, the samples were analyzed with a BD FACSCalibur system (Becton-Dickinson, San Jose, CA, USA).

### 2.6. Orthotopic Xenograft Model

Five-week-old male FoxnN.Cg/c nude mice (Orient Bio, Seongnam, Korea) were housed under pathogen-free conditions and received specific care based on research ethics outlined in a plan approved by the institutional animal care and use committee of Jeonbuk National University (approval number: JBNU 2021-032, date of approval: 25 February 2021). The mice acclimatized for one week were grouped randomly into five groups, with each group containing four mice. KHOS/NP cells were transfected with empty vectors, a plasmid for WT-CSNK2A1, plasmids for WT-CSNK2A1 and WT-SIRT6 (WT-CSNK2A1/WT-SIRT6), or a plasmid for SIRT6-S338A mutant. Thereafter, to establish the xenograft osteosarcoma model, 2 × 10^6^ KHOS/NP cells were injected into the bone marrow of the proximal tibia under anesthesia. Two weeks after tumor cell injection, the mice receiving doxorubicin according to the classification of the experimental groups were injected with doxorubicin [4 mg/kg in dimethyl sulfoxide (DMSO)] intraperitoneally once a week. Body weights and tumor sizes were measured every five days. The tumor sizes were estimated as length × width × height × 0.52 formula. According to the humane end-point, the mice were euthanized six weeks after doxorubicin treatment. Mice were sacrificed after anesthetizing with sodium pentobarbital. The tumors, lung, liver, and kidney were resected and evaluated histologically with hematoxylin and eosin staining. Animal experiments were approved by the institutional animal care and use committee of Jeonbuk National University (approval number: JBNU 2021-032).

### 2.7. Statistical Analysis

Immunohistochemical staining scores for CSNK2A1 were estimated by receiver operating characteristic curve analysis at the critical predictive point for the death of osteosarcoma patients [36]. The point with the highest area under the curve was determined as a cut-off point. The overall survival (OS) and relapse-free survival (RFS) were calculated to estimate the prognosis of osteosarcoma patients. The events of death for osteosarcoma patients occurred in OS analysis, and the duration of death was estimated from the time of operation to the time of last contact. In addition, the events of relapse or death of osteosarcoma patients occurred in RFS analysis, and the duration of death was calculated from the time of operation to the time of last contact. The prognostic survival analysis was calculated with univariate and multivariate Cox proportional hazards regression analysis, while the Kaplan–Meier survival analysis generated the survival curves. The Pearson’s chi-square test and Student’s *t*-test were used to calculate the relationship between the clinicopathological variables. All experiments were performed in triplicate and performed three times, with representative data presented. Statistical analysis was performed using the SPSS software (IBM, version 20.0, Armonk, NY, USA), and a *p* value less than 0.05 was considered significant.

## 3. Results

### 3.1. The Expression of CSNK2A1 Is Associated with Poor Prognosis of Osteosarcoma Patients

To evaluate the clinicopathological significance of the expression of CSNK2A1 and pSIRT6 in human osteosarcomas, we performed immunohistochemical staining for CSNK2A1 and pSIRT6. Representative images of the immunohistochemical expression pattern of CSNK2A1 and pSIRT6 are presented in Figure 1a. The positivity of the immunohistochemical expression of CSNK2A1 and pSIRT6 were determined with receiver operating characteristic curve analysis (Figure 1b). The cut-off point was determined at the point with the highest area under the curve to predict the death of osteosarcoma patients (Figure 1b). The cut-off points for both CSNK2A1 and pSIRT6 were twelve, and the cases with immunohistochemical staining scores equal to or greater than twelve were considered positive for CSNK2A1 immunostaining (Figure 1b). With these cut-off points, CSNK2A1-positivity was significantly associated with sex (*p* = 0.046), higher tumor stage (*p* = 0.006), higher T category (*p* = 0.028), higher M category (*p* = 0.047), latent distant metastasis (*p* = 0.025), and pSIRT6-positivity (*p* < 0.001) (Table 1). Positivity for pSIRT6 was significantly associated with sex (*p* = 0.014), tumor size (*p* = 0.031), higher T category (*p* = 0.011), and latent distant metastasis (*p* = 0.001) (Table 1). 

In univariate survival analysis, age (OS; *p* = 0.039, RFS; *p* = 0.018), tumor size (OS; *p* = 0.015, RFS; *p* = 0.019), tumor stage (OS; *p* = 0.015, RFS; *p* = 0.005), T category (OS; overall *p* = 0.059, RFS; overall *p* = 0.020), M category (OS; *p* = 0.007, RFS; *p* = 0.018), pSIRT6 expression (OS; *p* = 0.004, RFS; *p* = 0.001), and CSNK2A1 expression (OS; *p* = 0.002, RFS; *p* < 0.001) were significantly associated with OS or RFS (Table 2). Positivity of pSIRT6 expression predicted a 6.269-fold (95% confidence interval [95% CI]; 1.807–21.750) greater risk of death and a 7.783-fold (95% CI; 2.242–27.019) greater risk of relapse or death in osteosarcoma patients (Table 2). CSNK2A1-positivity predicted a 10.081-fold (95% CI; 2.307–44.054) greater risk of death and a 12.179-fold (95% CI; 2.777–53.407) greater risk of relapse or death in osteosarcoma patients (Table 2). The Kaplan–Meier survival curves for OS and RFS of CSNK2A1 and pSIRT6 expression are presented in Figure 1c. 

Multivariate analysis for OS and RFS was performed with the inclusion of age, tumor size, stage, T category, N category, M category, pSIRT6 expression, and CSNK2A1 expression. In multivariate analysis, CSNK2A1 expression was an independent indicator of OS and RFS (Table 3). CSNK2A1-positivity predicted a 10.147-fold (95% CI; 2.320–44.385, *p* = 0.002) greater risk of death and a 12.179-fold (95% CI; 2.777–53.407, *p* < 0.001) greater risk of relapse or death in osteosarcoma patients (Table 3). 

In the univariate analysis of 26 osteosarcoma patients who received adjuvant chemotherapy, the expression of CSNK2A1 (OS; *p* = 0.008, RFS; *p* = 0.003) and pSIRT6 (OS; *p* = 0.015, RFS; *p* = 0.008) were significantly associated with OS and RFS (Table 4) (Figure 2). In the multivariate analysis performed with the inclusion of age, tumor size, stage, T category, N category, M category, pSIRT6 expression, and CSNK2A1 expression, N category was an independent indicator of OS (*p* = 0.029), pSIRT6 expression was an independent indicator of OS (*p* = 0.011), and CSNK2A1 expression was an independent indicator of RFS (*p* = 0.003) (Table 4). Positivity of pSIRT6 expression predicted an 18.649-fold (95% CI; 1.949–178.412) greater risk in OS analysis, and CSNK2A1 expression predicted a 10.374–fold (95% CI; 2.244–47.968) greater risk in RFS analysis in osteosarcoma patients who received adjuvant chemotherapy (Table 4).

### 3.2. The Expression of CSNK2A1 Is Involved in the Resistance to the Anti-Proliferative Effect of Doxorubicin

In human osteosarcomas, especially in the patients who received adjuvant chemotherapy, the expression of CSNK2A1 was significantly associated with shorter survival. Therefore, we evaluated the effect of CSNK2A1 in doxorubicin cytotoxicity on osteosarcoma cells. In U2OS and KHOS/NP osteosarcoma cells, the overexpression of CSNK2A1 did not affect the proliferation of cells (Figure 3a,b). However, under doxorubicin treatment, the overexpression of WT-CSNK2A1 attenuated the anti-proliferative effect of doxorubicin (Figure 3a,b). Under doxorubicin treatment, the cellular proliferation of CSNK2A1-overexpressing cells was significantly higher than the cells transfected with empty vector (Figure 3a,b). In contrast, the knock-down of CSNK2A1 sensitized U2OS and KHOS/NP osteosarcoma cells to doxorubicin in the CCK8 assay and colony-forming assay (Figure 3c,d). The proliferation of osteosarcoma cells that had a knock-down of CSNK2A1 was significantly lower compared with cells transfected with empty vector under treatment with doxorubicin (Figure 3c,d).

### 3.3. CSNK2A1 Induces Resistance to Doxorubicin-Mediated Apoptosis of Osteosarcoma Cells

When the U2OS and KHOS/NP osteosarcoma cells were treated with doxorubicin, the expression of cleaved PARP1, cleaved caspase 3, and BAX increased, and the expression of BCL2 decreased (Figure 4). Under doxorubicin treatment of U2OS and KHOS/NP osteosarcoma cells, the expression levels of cleaved PARP1, cleaved caspase 3, and BAX increased, and the expression of BCL2 decreased with the knock-down of CSNK2A1 (Figure 4). Overexpression of CSNK2A1 decreased the expression levels of cleaved PARP1, cleaved caspase 3, and BAX and increased the expression of BCL2 under doxorubicin treatment (Figure 4). In flow-cytometry apoptotic analysis, the apoptosis of U2OS and KHOS/NP cells significantly increased with the knock-down of CSNK2A1 but significantly decreased with the overexpression of CSNK2A1 compared with controls under treatment with doxorubicin (Figure 5).

### 3.4. CSNK2A1-Mediated Resistance to Doxorubicin Is Associated with SIRT6 Phosphorylation-Mediated Activation of the DNA Damage Repair Pathway

When U2OS and KHOS/NP osteosarcoma cells were treated with doxorubicin, the expression of pSIRT6, pATM, pChk2, and γH2AX were increased (Figure 6). In addition, under treatment with doxorubicin in U2OS and KHOS/NP cells, the knock-down of CSNK2A1 decreased the expression of pSIRT6, pATM, and pChk2, and the overexpression of CSNK2A1 increased the expression of pSIRT6, pATM, and pChk2 (Figure 6). The expression of γH2AX increased with the knock-down of CSNK2A1 and decreased with the overexpression of CSNK2A1 (Figure 6). Based on the role of CSNK2A1 on the phosphorylation of SIRT6 on Ser338 [10] and the role of SIRT6 on the induction of the DNA damage repair pathway [21], we evaluated the effect of a mutation of the Ser338 phosphorylation site of SIRT6 on the CSNK2A1-mediated activation of the DNA damage repair pathway triggered by treatment with doxorubicin. The overexpression of WT-CSNK2A1 or WT-CSNK2A1/WT-SIRT6 increased the expression of pSIRT6, pATM, and pChk2 and decreased the expression of γH2AX with doxorubicin treatment in KHOS/NP cells (Figure 7a). However, despite the overexpression of CSNK2A1, the transfection of the SIRT6-S338A mutant decreased the expression of pSIRT6, pATM, and pChk2, which were induced by doxorubicin and the overexpression of CSNK2A1 (Figure 7a). In CCK8 and colony-forming assays, cells overexpressing WT-CSNK2A1 or WT-CSNK2A1/WT-SIRT6 were resistant to doxorubicin, and the resistance to doxorubicin was attenuated with the transfection of the SIRT6-S338A mutant (Figure 7b,c). The number of cells and colonies were significantly decreased in the cells transfected with the SIRT6-S338A mutant compared with the cells overexpressing WT-CSNK2A1 or WT-CSNK2A1/WT-SIRT6 under treatment with doxorubicin (Figure 7b,c). 

In an orthotopic xenograft model, the growth of tumors in vivo was significantly inhibited with doxorubicin treatment (4 mg/kg, once a week, intraperitoneal injection) (Figure 8a,b). However, the anti-tumor effect of doxorubicin was attenuated via the overexpression of WT-CSNK2A1 or WT-CSNK2A1/WT-SIRT6 (Figure 8a,b). The growth of tumors in vivo was significantly higher with the overexpression of WT-CSNK2A1 or WT-CSNK2A1/WT-SIRT6, compared with transfection of empty vector under treatment with doxorubicin (Figure 8a,b). The resistance to doxorubicin induced by the overexpression of WT-CSNK2A1 and WT-SIRT6 was attenuated by a mutation of SIRT6 on Ser338. In vivo tumor growth of KHOS/NP cells was also significantly decreased with the transfection of the SIRT6-S338A mutant compared with the overexpression of WT-CSNK2A1 or WT-CSNK2A1/WT-SIRT6 (Figure 8a–c). Furthermore, pulmonary metastasis was significantly decreased in the KHOS/NP cells transfected SIRT6-S338A mutant (number of metastatic nodule per mouse: mean ± standard deviation, 0 ± 0), compared with cells with induced overexpression of WT-CSNK2A1 (number of metastatic nodule per mouse: mean ± standard deviation, 3.0 ± 1.2) or cells induced to overexpress WT-CSNK2A1/WT-SIRT6 (number of metastatic nodule per mouse: mean ± standard deviation, 3.5 ± 2.4) (Figure 8d). There were no metastases on the liver or kidney in all experimental groups.

### 3.5. Inhibition of CSNK2A1 with Emodin Potentiates the Cytotoxic Effects of Doxorubicin

Based on the role of CSNK2A1 in resistance to doxorubicin, we evaluated the effects of emodin, a CSNK2A1 inhibitor, on osteosarcoma cells. Treatment with emodin inhibited the proliferation of U2OS and KHOS/NP cells in a dose- and time-dependent manner (Figure 9a). As shown in Figure 3, the knock-down or overexpression of CSNK2A1 did not affect the proliferation of osteosarcoma cells without doxorubicin treatment. However, in contrast, emodin showed significant anti-proliferative effects with 24 h of treatment at a 0.4 mM concentration (Figure 9a). Moreover, similar to when cells underwent a knock-down of CSNK2A1, 0.5 mM emodin treatment potentiated the cytotoxic effects of doxorubicin (Figure 9b,c). Co-treatment with emodin and doxorubicin significantly inhibited the proliferation of U2OS and KHOS/NP cells compared with treatment with either doxorubicin or emodin alone (Figure 9b,c).

## 4. Discussion

In this study, we demonstrated that the expression of CSNK2A1 was significantly associated with higher tumor stage and latent distant metastasis of osteosarcoma. In addition, CSNK2A1-positivity predicted shorter OS and RFS in osteosarcoma patients. Furthermore, in the sub-population of osteosarcoma patients who received postoperative chemotherapy, CSNK2A1 expression was significantly associated with shorter survival of patients. Consistently, the expression of CSNK2A1 was elevated in cancers compared with non-cancerous tissues in the colorectum [4], head and neck [6], kidney [7,8], prostate [9], and stomach [5]. In addition, the prognostic significance of CSNK2A1 in human cancer patients has been reported in various types of cancers, such as breast cancer [10], clear cell renal cell carcinoma [8,37], colon cancer [4,38], gastric cancer [5], and prostatic cancer [9]. In clear cell renal cell carcinoma, higher expression of CSNK2A1 mRNA was associated with higher tumor stage, higher nuclear grade, presence of distant metastasis, and shorter OS [8]. In breast and gastric carcinomas, immunohistochemical expression of CSNK2A1 was an independent indicator of shorter OS and RFS in cancer patients [5,10]. Specifically, with regards to the subcellular localization of CSNK2A1, its high expression in the nucleoli of breast cancer cells predicted shorter survival of patients in univariate and multivariate analysis [39]. These findings suggest that CSNK2A1 is important in the progression of human cancers and might be employed as a prognostic indicator of human cancers. 

The molecular mechanisms and roles of CSNK2A1 in tumorigenesis have been extensively investigated, and it has been shown that CSNK2A1 is involved in tumorigenesis by its roles as a kinase to activate oncogenic molecules and to stimulate degradation of tumor suppressors [3,12,40,41]. Consequently, CSNK2A1 promotes the proliferation and invasiveness of cancer cells by activating cell-cycle progression and the epithelial-to-mesenchymal transition (EMT) [4,5,10,13]. CSNK2A1 engages the MYC [11], Akt [13,14], and NFκB [12] pathways to stimulate the proliferation of cancer cells. In myeloid neoplasms, the depletion of CSNK2A1 inhibited cell-cycle progression by elevating P27 [42]. CSNK2A1 also plays an important role in cancer invasiveness by activating the EMT pathway in colorectal cancer [4] and breast cancer [10]. In breast cancer cells, CSNK2A1 is involved in the EMT by regulating the Wnt/β-catenin pathway [10]. In gastric cancer cells, CSNK2A1 is involved in cancer progression by stimulating proliferation and invasiveness through the phosphorylation of DBC1/CCAR2 [5] and the PI3K-Akt-mTOR pathway [43]. In addition, CSNK2A1 stimulates the degradation of the tumor suppressors PTEN [44] and P53 [40]. Moreover, CSNK2A1 renders cancer cells resistant to apoptotic stresses by activating BCL2 and suppressing PML, FOXO3, and PARP1 [3,10,41]. However, the effect of CSNK2A1 on the proliferation of osteosarcoma cells has been controversially reported. A previous report showed that the knock-down or inhibition of CSNK2A1 inhibits the proliferation of osteosarcoma cells [45]; however, in this study, the knock-down or overexpression of CSNK2A1 did not influence the proliferation of U2OS and KHOS/NP cells. Therefore, a careful approach is needed to evaluate the possibility of CSNK2A1 as a potential therapeutic target of osteosarcoma based on the effect of CSNK2A1 on the proliferation of osteosarcoma cells. 

An interesting finding of this study is that the CSNK2A1-mediated phosphorylation of SIRT6 is important in the treatment efficacy of doxorubicin in osteosarcoma cells. Specifically, the phosphorylation of SIRT6 on Ser338 was critical in CSNK2A1-mediated resistance to doxorubicin in vitro and in vivo. In addition, the higher expression of pSIRT6 was associated with shorter survival in osteosarcoma patients. Concerning these results, when considering the molecular mechanism of CSNK2A1 in cancer to be used as a therapeutic target, the molecular network related to CSNK2A1 is complex, because CSNK2A1 might also be protective for molecules that might have tumor-suppressive roles [10,18]. CSNK2A1 phosphorylates SIRT6 on Ser338 [10]. However, there are controversial reports on the role of SIRT6 in tumorigenesis. With respect to a tumor-suppressive role of SIRT6, the loss of SIRT6 stimulated the progression of hepatocellular carcinoma [46], and lower expression of SIRT6 was associated with poor prognosis in breast cancer patients [23]. Furthermore, the role of SIRT6 on the induction of the DNA damage repair pathway strongly supports its role in the prevention of cancer development [47,48]. However, despite these reports, there are also reports that the higher expression of SIRT6 is closely associated with shorter survival in cancer patients, and SIRT6 is involved in the stimulation of molecules closely associated with cancer progression [10,20,21,27,49]. In addition, controversially, there were opposing reports on the role of SIRT6 in the same cancer type. The effect of SIRT6 on the proliferation of MDA-MB-231 breast cancer cells was the opposite in two studies [10,23]. In a study using osteosarcoma cells, the overexpression of SIRT6 inhibited the proliferation and invasiveness of SAOS2 and MG-63 osteosarcoma cells [24]; however, the study used U2OS and KHOS/NP osteosarcoma cells, and SIRT6 did not affect the proliferation of osteosarcoma cells despite chemo-sensitization with the knock-down of SIRT6 [21]. Therefore, we conducted a more precise evaluation of the role of CSNK2A1-SIRT6 in human cancer, especially in osteosarcoma, and we showed that the phosphorylation status of SIRT6 on Ser338 is important in CSNK2A1-mediated chemoresistance in osteosarcoma cells. 

In our results, the knock-down of CSNK2A1 or the inhibition of CSNK2A1 with emodin sensitized osteosarcoma cells to doxorubicin. Similarly, the inhibition of CSNK2A1 with CX-4945 suppressed the proliferation of osteosarcoma cells [45] and Caki-2 clear cell renal cell carcinoma cells [37]. In addition, emodin potentiated the anti-cancer effect of cisplatin by inhibiting P-glycoprotein expression in non-small cell lung cancer cells [30]. In line with our results, the chemo-resistant phenotype of cancer cells was associated with the higher expression of CK2, and the inhibition of CK2 inhibited the proliferation of cancer cell lines with chemoresistance [50]. The mechanism by which CSNK2A1 renders a resistance phenotype to cancer cells has been suggested to be by suppressing the apoptotic signaling of cancer cells [3,10,41]. Concerning this mechanism, our results indicate that CSNK2A1 induces DNA damage repair pathway by the activation of SIRT6 through the phosphorylation of SIRT6 on Ser338. As we previously reported, CSNK2A1 phosphorylates SIRT6 on Ser338 among the four phosphorylation sites of SIRT6 [10], and expression of CSNK2A1 was associated with phosphorylation of SIRT6 on Ser338 on U2OS and KHOS/NP osteosarcoma cells. In addition, based on the role of SIRT6 in DNA damage repair, the higher expression of SIRT6 was important in resistance to doxorubicin through the activation of the DNA damage repair pathway [21]. Similarly, the inhibition of SIRT6 sensitized diffuse large B-cell lymphoma cells on doxorubicin or bendamustine by suppressing the DNA damage repair pathway [49]. Therefore, in conjunction with previous reports and this study, it is suggested that the CSNK2A1-mediated phosphorylation of SIRT6 on Ser338 is important in rescuing injured cancer cells by repairing DNA damage induced by anti-cancer therapeutics. Therefore, both blocking CSNK2A1 and SIRT6 or the direct suppression of the DNA damage repair pathway might be promising therapeutic strategies to overcome cancer resistance to conventional genotoxic anti-cancer therapeutics. The FDA approves the use of inhibitors of the DNA damage repair pathway, such as PARP inhibitors, in cancers of the ovary and prostate (OncoKB database, https://www.oncokb.org, accessed on 10 June 2021). In osteosarcoma cells, the PARP inhibitor olaparib also potentiated the cytotoxic effect of doxorubicin [27]. Therefore, when considering CSNK2A1-mediated resistance to doxorubicin as it relates to the DNA damage repair response, the use of PARP inhibitors also might be a potential therapeutic application in the treatment of poor prognostic osteosarcoma highly expressing CSNK2A1. However, there is a limitation to the use of inhibitors of PARP and SIRT6 in sarcomas, including osteosarcoma. In this study, we have shown that the CSNK2A1 inhibitor emodin synergistically potentiates the cytotoxic effect of doxorubicin in osteosarcoma cells. Several additional chemicals inhibiting CSNK2A1 are under evaluation in human cancers [3,29,45,50]. Therefore, blocking CSNK2A1 might be helpful in achieving therapeutic efficacy in osteosarcoma highly expressing CSNK2A1 and/or SIRT6. However, this study has the limitation that we have used just one CSNK2A1 inhibitor and did not evaluate the effectiveness of emodin in vivo. Therefore, additional studies evaluating the effectiveness of various inhibitors of CSNK2A1 in osteosarcoma are needed.

## 5. Conclusions

In this study, we present that the expression of CSNK2A1 and pSIRT6 might be used as important prognostic indicators of osteosarcoma patients, especially for patients who received postoperative chemotherapy. Furthermore, we have shown that CSNK2A1 induces resistance to doxorubicin through SIRT6 phosphorylation-mediated activation of the DNA damage repair pathway in vitro and in vivo. Therefore, our results suggest that the blocking of the CSNK2A1-SIRT6-DNA damage repair pathway might be a promising therapeutic strategy for osteosarcoma patients, especially for the poor prognostic subpopulation of patients who have tumors highly expressing CSNK2A1.

## Figures and Tables

**Figure 1 cells-10-01770-f001:**
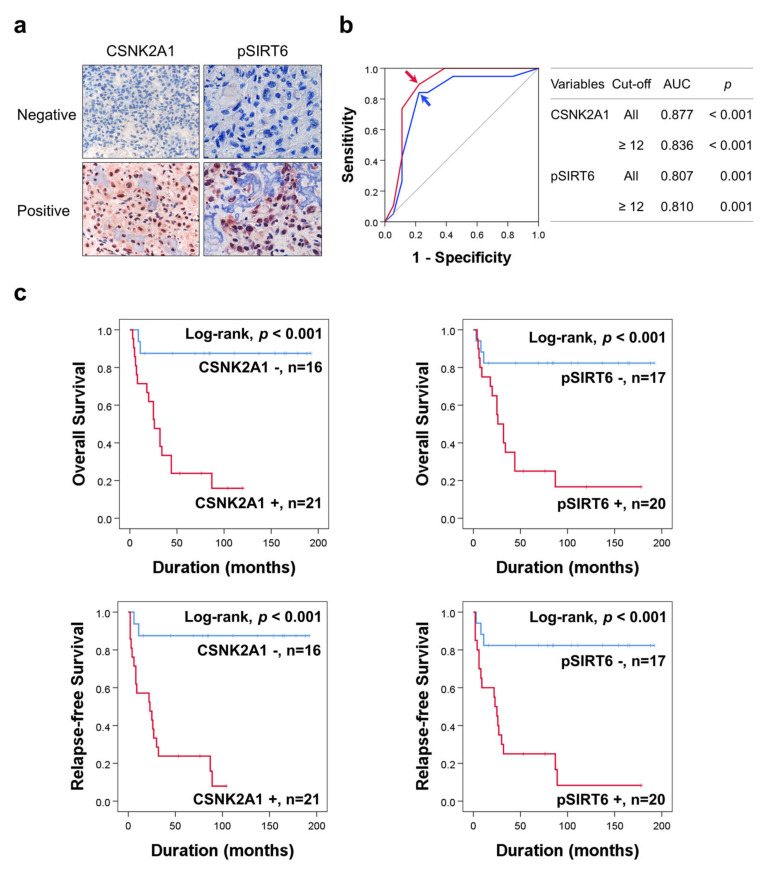
Immunohistochemical expression of CSNK2A1 and pSIRT6 in human osteosarcoma and statistical and survival analysis in osteosarcoma. (**a**) Negative and positive examples of immunohistochemical expression of CSNK2A1 and pSIRT6 in human osteosarcoma tissue. (**b**) The cut-off points of CSNK2A1 and pSIRT6 expression were determined using receiver operating characteristic curve analysis. The cut-off points of the immunohistochemical staining score for CSNK2A1 and pSIRT6 were twelve for both marker, and were determined at the point with the highest area under the curve (AUC). The red arrow (CSNK2A1) and blue arrow (pSIRT6) indicate the cut-off points. (**c**) Kaplan–Meier survival curves for overall survival and relapse-free survival according to the expression of CSNK2A1 and pSIRT6 in 37 osteosarcoma patients.

**Figure 2 cells-10-01770-f002:**
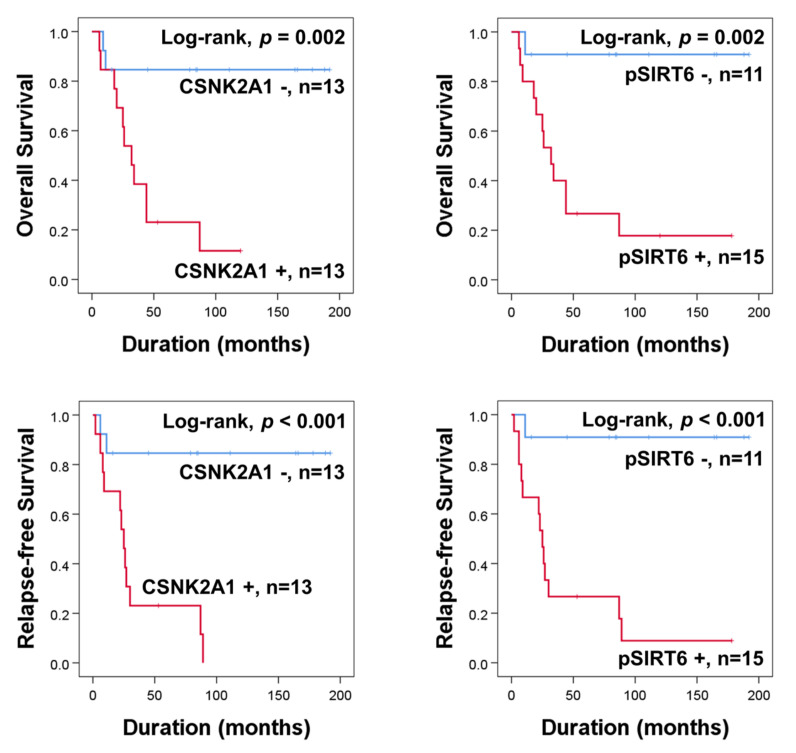
Kaplan–Meier survival curves for overall survival and relapse-free survival according to expression of CSNK2A1 and pSIRT6 in 26 osteosarcoma patients who received adjuvant chemotherapy.

**Figure 3 cells-10-01770-f003:**
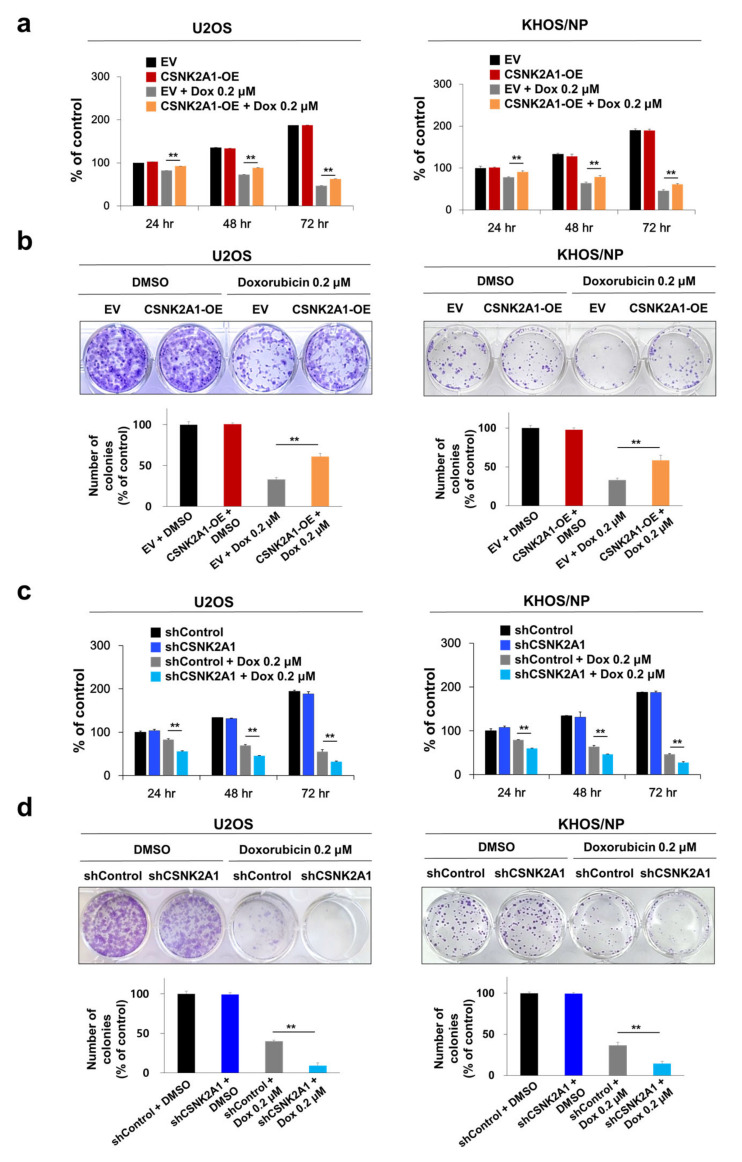
CSNK2A1 involvement in the effects of doxorubicin on the proliferation of osteosarcoma cells. (**a**) U2OS and KHOS/NP osteosarcoma cells were transfected with empty vector or wild-type CSNK2A1 and treated with doxorubicin (0.2 μM) and then measured after 24, 48, and 72 h for proliferation with a CCK8 proliferation assay. (**b**) For the colony-forming assay, U2OS (1 × 10^3^) and KHOS/NP (1 × 10^3^) cells transfected with empty vector or wild-type CSNK2A1 were treated with DMSO or 0.2 μM doxorubicin in 24-well culture plates. The cells were grown for ten days. Clono-Counter software was used in the quantification of the number of colonies. (**c**) U2OS and KHOS/NP osteosarcoma cells were transfected with control shRNA or shRNA for CSNK2A1 and treated with 0.2 μM doxorubicin and then measured after 24, 48, and 72 h for proliferation with a CCK8 proliferation assay. (**d**) For the colony-forming assay, U2OS (1 × 10^3^) and KHOS/NP (1 × 10^3^) cells transfected with control shRNA or shRNA for CSNK2A1 were treated with DMSO or 0.2 μM doxorubicin in 24-well culture plates. The cells were grown for ten days. Clono-Counter software was used in the quantification of the number of colonies. **; *p* < 0.001.

**Figure 4 cells-10-01770-f004:**
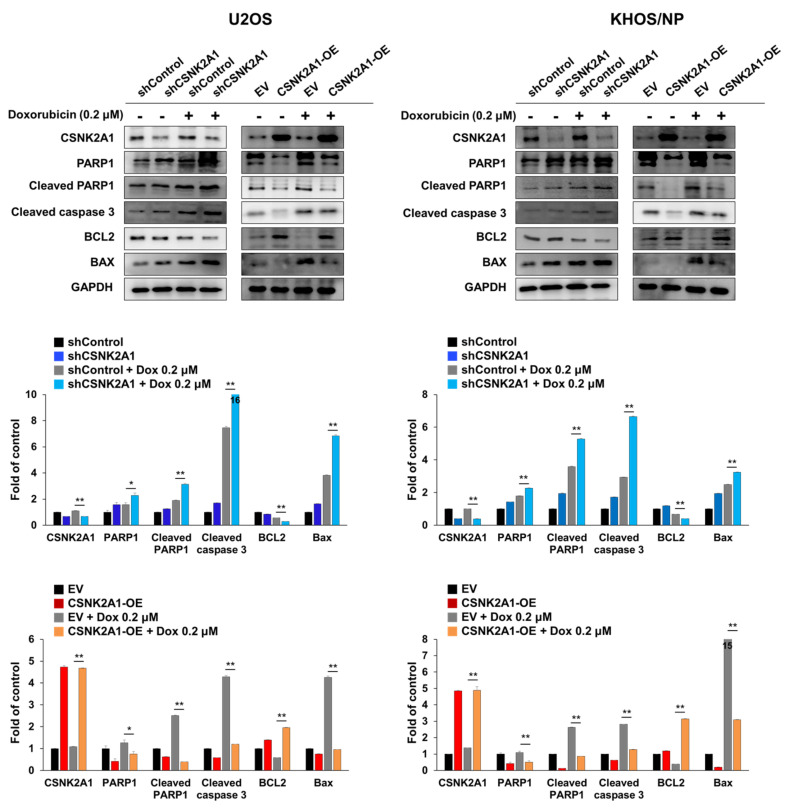
CSNK2A1 involvement in the apoptotic effects of doxorubicin in osteosarcoma cells with western blot analysis. U2OS and KHOS/NP osteosarcoma cells were transfected with control shRNA, shRNA for CSNK2A1, empty vector, or wild-type CSNK2A1 and treated with 0.2 μM doxorubicin. The expression patterns of CSNK2A1, PARP1, cleaved PARP1, cleaved caspase 3, BCL2, BAX, and GAPDH were determined via Western blot. The density of the Western blot bands was measured in triplicate by using ImageJ software. *; *p* < 0.05, **; *p* < 0.001.

**Figure 5 cells-10-01770-f005:**
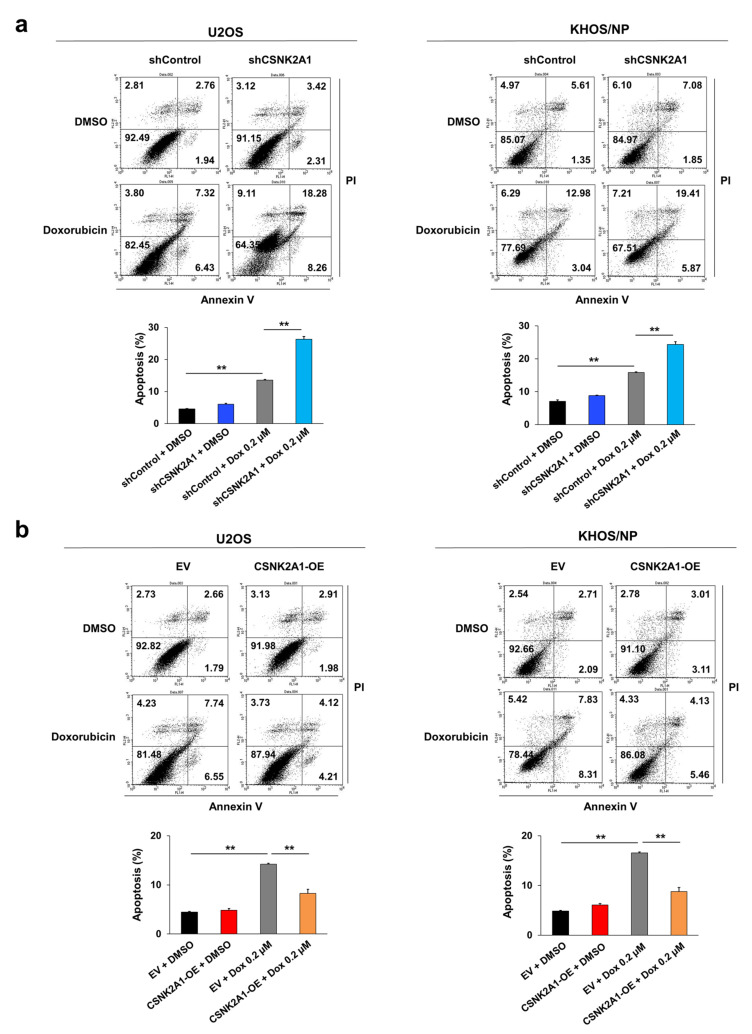
CSNK2A1 involvement in the apoptotic effects of doxorubicin in osteosarcoma cells with flow cytometry analysis. (**a**) U2OS and KHOS/NP osteosarcoma cells were transfected with control shRNA or shRNA for CSNK2A1 and treated with 0.2 μM doxorubicin for 24 h. (**b**) U2OS and KHOS/NP osteosarcoma cells were transfected with empty vector or wild-type CSNK2A1 and treated with 0.2 μM doxorubicin for 24 h. The apoptosis of cells was evaluated via flow-cytometry analysis with staining with propidium iodide and for annexin V. **; *p* < 0.001.

**Figure 6 cells-10-01770-f006:**
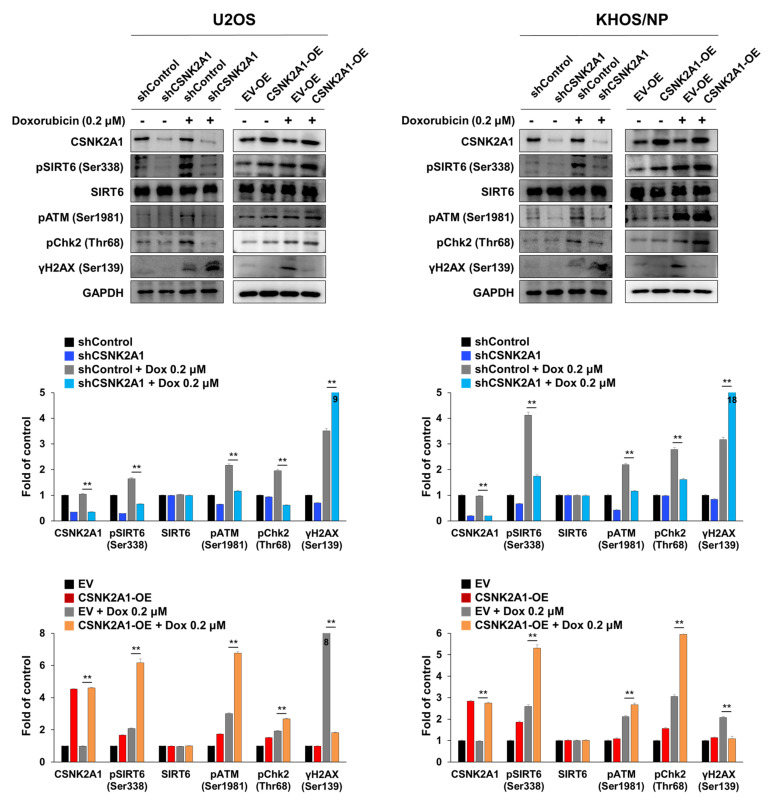
CSNK2A1 involvement in doxorubicin resistance through the SIRT6-mediated activation of the DNA damage repair pathway, shown using Western blot analysis. U2OS and KHOS/NP cells were transfected with control shRNA, shRNA for CSNK2A1, empty vector, or WT-CSNK2A1 and treated with DMSO or 0.2 μM doxorubicin. Western blots for CSNK2A1, SIRT6, phosphorylated SIRT6 (pSIRT6, Ser338), phosphorylated ATM (pATM), phosphorylated Chk2 (pChk2), phosphorylated H2AX (γH2AX, ser138), and GAPDH were performed, and the density of the bands was measured in triplicate by using ImageJ software. **; *p* < 0.001.

**Figure 7 cells-10-01770-f007:**
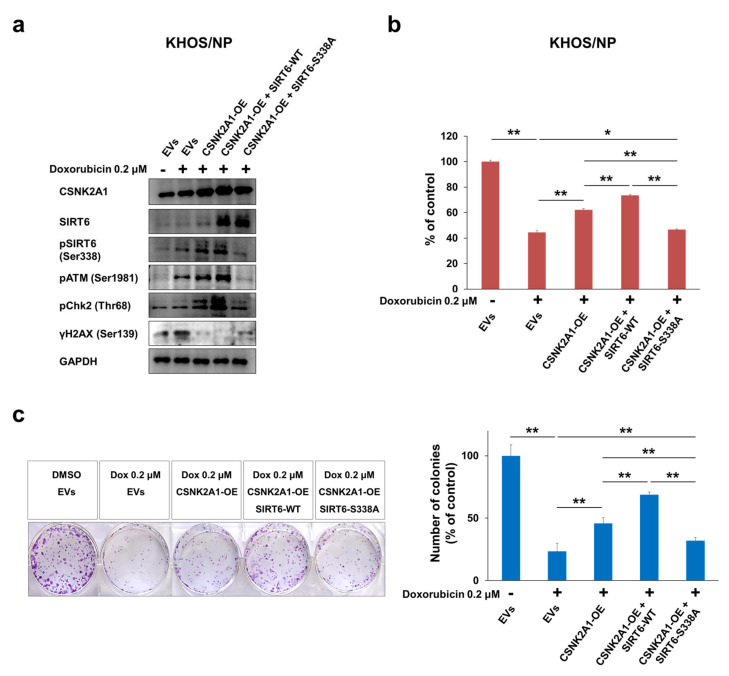
CSNK2A1 involvement in doxorubicin resistance through SIRT6-phosphorylation-mediated activation of the DNA damage repair pathway. (**a**) KHOS/NP cells transfected with empty vectors (EVs), WT-CSNK2A1, WT-CSNK2A1 and WT-SIRT6 (SIRT6-WT; pFLAG2_SIRT6) or WT-CSNK2A1 and mutant SIRT6 (pSIRT6-S338A; pFLAG2_SIRT6_S338A) were treated with 0.2 μM doxorubicin, and Western blots were performed for CSNK2A1, SIRT6, phosphorylated SIRT6 (pSIRT6, Ser338), phosphorylated ATM (pATM), phosphorylated Chk2 (pChk2), phosphorylated H2AX (γH2AX, ser138), and GAPDH. (**b**) The proliferation of transfected KHOS/NP (1 × 10^3^) cells with indicated vectors was measured 24 h after seeding with a CCK8 proliferation assay. (**c**) Colony-forming assays were performed by seeding transfected KHOS/NP (2 × 10^3^) cells with the indicated vectors after treatment with DMSO or 0.2 μM doxorubicin in 6-well culture plates. The cells were grown for ten days. Clono-Counter software was used in the quantification of the number of colonies. EVs; the cells transfected with both empty vector for CSNK2A1 and empty vector for SIRT6. *; *p* < 0.05, **; *p* < 0.001.

**Figure 8 cells-10-01770-f008:**
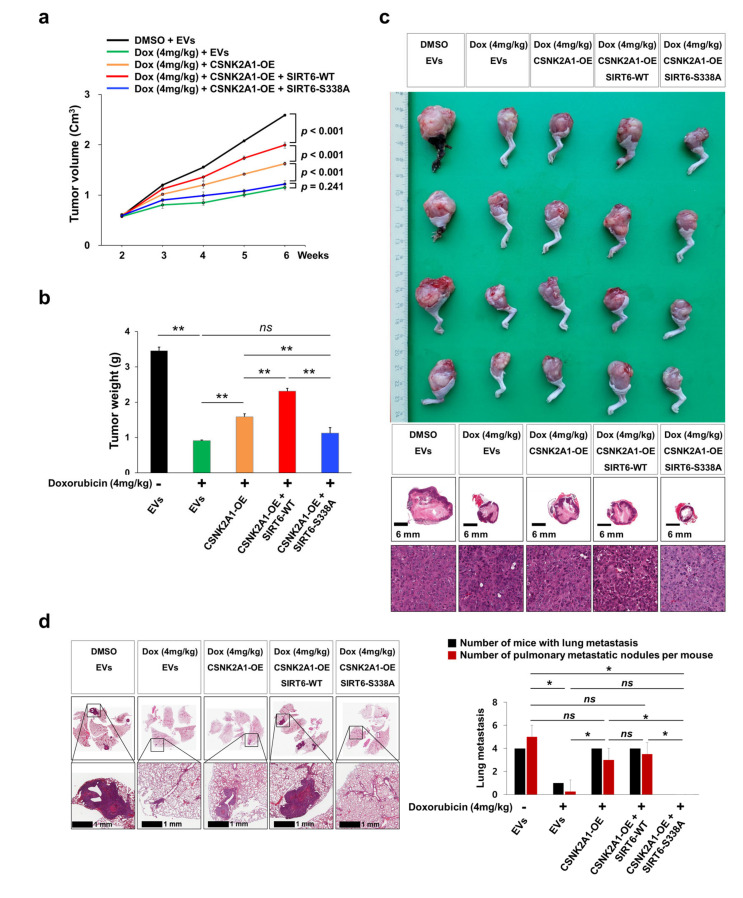
CSNK2A1-mediated resistance to doxorubicin in osteosarcoma cells is attenuated by the mutation of SIRT6 in vivo. (**a**) KHOS/NP (2 × 10^6^) cells were transfected with empty vectors (EVs), WT-CSNK2A1, WT-CSNK2A1, and WT-SIRT6 (SIRT6-WT; pFLAG2_SIRT6) or WT-CSNK2A1 and mutant SIRT6 (pSIRT6-S338A; pFLAG2_SIRT6_S338A) and injected into the bone marrow of the proximal tibia under anesthesia. Two weeks after tumor implantation, doxorubicin (4 mg/kg in DMSO) was injected intraperitoneally once a week, and the volume of the tumor was measured every week. The tumor volumes were calculated as length × width × height × 0.52. (**b**) Six weeks after tumor injection, tumor weight was measured after the euthanization of mice. (**c**) Macroscopic and microscopic images of orthotopic xenografted tumors. (**d**) Microscopic findings of pulmonary metastatic nodules at six weeks after the injection of KHOS/NP cells. The data for the number of mice with pulmonary metastasis and the number of pulmonary metastatic nodules per mouse is presented as a graph. The number of metastatic nodules was counted under microscopy. *ns*; not significant, *; *p* < 0.05, **; *p* < 0.001.

**Figure 9 cells-10-01770-f009:**
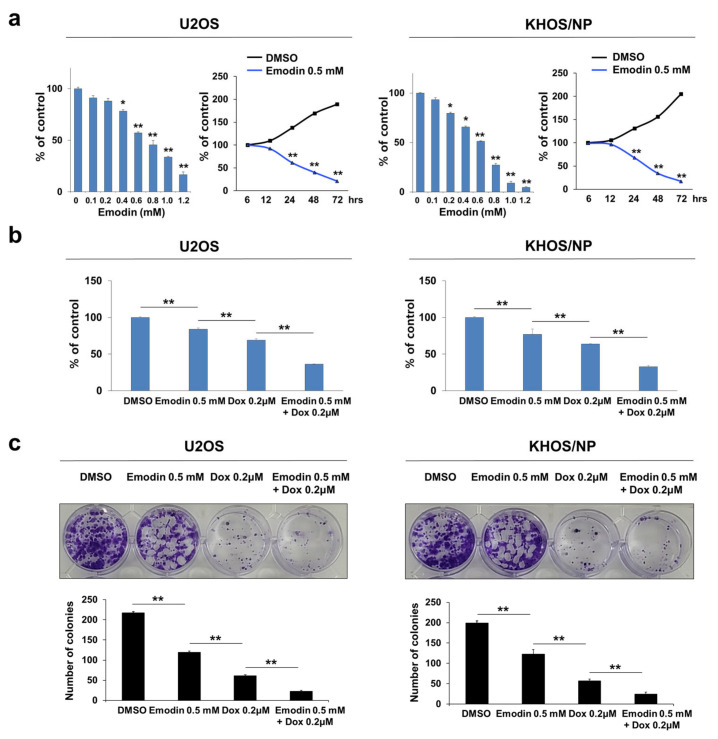
Co-treatment with CSNK2A1-inhibitor emodin potentiated the anti-proliferative effect of doxorubicin in osteosarcoma cells. (**a**) U2OS and KHOS/NP cells were treated with DMSO or 0.1~1.2 mM emodin, a CSNK2A1 inhibitor, for 24 h. In addition, U2OS and KHOS/NP cells were treated with DMSO or 0.5 mM emodin for 6, 12, 24, 48, and 72 h. The proliferation of cells was evaluated with a CCK8 proliferation assay. (**b**) U2OS and KHOS/NP cells treated with DMSO, 0.5 mM emodin, and 0.2 μM doxorubicin or 0.5 mM emodin and 0.2 μM doxorubicin for 24 h and evaluated with a CCK8 proliferation assay. (**c**) Colony-forming assays were performed with U2OS (1 × 10^3^) and KHOS/NP (1 × 10^3^) cells treated with DMSO, 0.5 mM emodin, and 0.2 μM doxorubicin or 0.5 mM emodin and 0.2 μM doxorubicin in 24-well culture plates. Clono-Counter software was used in the quantification of the number of colonies. *; *p* < 0.05, **; *p* < 0.001.

**Table 1 cells-10-01770-t001:** The clinicopathological characteristics of 37 osteosarcomas and their association with the immunohistochemical expression of CSNK2A1.

Characteristics		No.	CSNK2A1		pSIRT6	
			Positive	*p*	Positive	*p*
Age, years	<30	24	11 (46%)	0.068	12 (50%)	0.501
	≥30	13	10 (77%)		8 (62%)	
Sex	Male	25	17 (68%)	0.046	17 (68%)	0.014
	Female	12	4 (33%)		3 (25%)	
Tumor size	≤8 cm	19	8 (42%)	0.065	7 (37%)	0.031
	>8 cm	18	13 (72%)		13 (72%)	
Stage	I & II	26	11 (42%)	0.006	12 (46%)	0.138
	III & IV	11	10 (91%)		8 (73%)	
T category	1	17	6 (35%)	0.028	5 (29%)	0.011
	2	16	11 (69%)		11 (69%)	
	3 and 4	4	4 (100%)		4 (100%)	
N category	N0	34	19 (56%)	0.718	19 (56%)	0.452
	N1	3	2 (67%)		1 (33%)	
M category	M0	29	14 (48%)	0.047	15 (52%)	0.588
	M1	8	7 (88%)		5 (63%)	
Latent distant metastasis	Absence	28	13 (46%)	0.025	11 (39%)	0.001
	Presence	9	8 (89%)		9 (100%)	
pSIRT6	Negative	17	3 (18%)	<0.001		
	Positive	20	18 990%)			

**Table 2 cells-10-01770-t002:** Univariate analysis with Cox proportional hazards regression analysis for the survival of 37 osteosarcoma patients.

Characteristics	No.	OS		RFS	
		HR (95% CI)	*p*	HR (95% CI)	*p*
Age, years, ≥30 (vs. <30)	13/37	2.599 (1.050–6.433)	0.039	2.902 (1.201–7.014)	0.018
Sex, male (vs. female)	25/37	0.825 (0.294–2.312)	0.714	0.657 (0.235–1.836)	0.423
Tumor size, ≥8 cm (vs. <8 cm)	18/37	3.359 (1.263–8.933)	0.015	3.076 (1.207–7.838)	0.019
Stage, III & IV (vs. I & II)	11/37	3.161 (1.255–7.961)	0.015	3.647 (1.472–9.034)	0.005
T category, 1	17/37	1	0.059	1	0.020
2	16/37	3.335 (1.150–9.668)	0.027	3.455 (1.188–10.043)	0.023
3 and 4	4/37	3.946 (0.938–16.606)	0.061	5.895 (1.559–22.287)	0.009
N category, N1 (vs. N0)	3/37	4.841 (0.957–24.486)	0.057	2.800 (0.599–13.095)	0.191
M category, M1 (vs. M0)	8/37	3.973 (1.464–10.784)	0.007	3.349 (1.229–9.125)	0.018
pSIRT6, positive (vs. negative)	20/37	6.269 (1.807–21.750)	0.004	7.783 (2.242–27.019)	0.001
CSNK2A1, positive (vs. negative)	21/37	10.081 (2.307–44.054)	0.002	12.179 (2.777–53.407)	<0.001

OS, overall survival; RFS, relapse-free survival; HR, hazard ratio; 95% CI, 95% confidence interval.

**Table 3 cells-10-01770-t003:** Multivariate Cox proportional hazards regression analysis for the survival of 37 osteosarcoma patients.

Characteristics	OS		RFS	
	HR (95% CI)	*p*	HR (95% CI)	*p*
N category, N1 (vs. N0)	5.099 (0.945–27.506)	0.058		
CSNK2A1, positive (vs. negative)	10.147 (2.320–44.385)	0.002	12.179 (2.777–53.407)	<0.001

OS, overall survival; RFS, relapse-free survival; HR, hazard ratio; 95% CI, 95% confidence interval. The variables included in multivariate analysis were age, tumor size, stage, T category, N category, M category, pSIRT6 expression, and CSNK2A1 expression.

**Table 4 cells-10-01770-t004:** Univariate and multivariate Cox proportional hazards regression analysis of the survival of 26 osteosarcoma patients who received adjuvant chemotherapy.

Characteristics		No.	OS		RFS	
			HR (95% CI)	*p*	HR (95% CI)	*p*
Univariate analysis						
	CK2α, positive (vs. negative)	13/26	7.741 (1.693–35.408)	0.008	10.374 (2.224–47.968)	0.003
	pSIRT6, positive (vs. negative)	15/26	12.682 (1.639–98.119)	0.015	15.709 (2.035–121.271)	0.008
Multivariate analysis						
	N category, N1 (vs. N0)		29.727 (1.425–620.194)	0.029		
	pSIRT6, positive (vs. negative)		18.649 (1.949–178.412)	0.011		
	CK2α, positive (vs. negative)				10.374 (2.244–47.968)	0.003

OS, overall survival; RFS, relapse-free survival; HR, hazard ratio; 95% CI, 95% confidence interval. The variables included in multivariate analysis were age, tumor size, stage, T category, N category, M category, pSIRT6 expression, and CSNK2A1 expression.

## Data Availability

The datasets used and/or analyzed during the current study are available from the corresponding author upon reasonable request.

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
