# Peer review of "CK2α/CSNK2A1 Induces Resistance to Doxorubicin through SIRT6-Mediated Activation of the DNA Damage Repair Pathway"

_cells, 2021, doi:10.3390/cells10071770_

Round 1

Reviewer 1 Report

The manuscript by Hussein et al. reports findings on the role of the CSNK2A1-SIRT6 axis in the chemotherapy response of osteosarcoma (OS) cells. The  constitutively active Ser/Thr protein kinase CSNK2A1 phosphorylates hundreds of substrates and controls several signaling pathways. It has been implicated, among many other conditions, in cancer, where it regulates practically all malignant hallmarks. CSNK2A1-mediated SIRT6 phosphorylation has also been reported in the progression of other cancers. The novelty of the current paper is that it systematically evaluates the role of the CSNK2A1-SIRT6 axis in OS and unequivocally demonstrates that this pathway determines the outcome of  doxorubicine-treatment of OS cells. The findings suggest that CSNK2A1 might be used as a prognostic indicator in osteosarcoma patients, especially in patients who received postoperative chemotherapy. Furthermore, the authors have shown that CSNK2A1 induces resistance to DOX through SIRT6 phosphorylation-mediated activation of the DNA damage repair pathway in vitro and in vivo.

The paper is well written and easy to follow. The figures were carefully designed although the figure texts are often difficult to read due to very small font sizes (e.g. on figures 3, 5, 6, 7). The conclusions are supported by the data.

Comments

  1. The authors chose DOX as a chemotheraputic drug to study in cell culture experiments. It is recommended to justify this choice in the paper. Was DOX a part of the chemotherapy given to patients enrolled in the study? What else was included in the complex chemotherapy of the patients?
  2. All Western blots should be evaluated by densitometry followed by statistical analysis.  In the current version of the MS, only representative WB images are presented (Fig 4A and 5A).
  3. Since the MS contains very large amounts of data, the reviewer is hesitant to suggest further experiments. However, measuring one more parameter [i.e. poly(ADP-ribose) polymer ] would add important mechanistic information to the role of the CSNK2A1-SIRT6 axis in the DNA damage response. Showing PAR polymer levels after CSNK2A1 overexpression/silencing would be useful especially in light of the fact that SIRT6 is known to activate PARP1.
  4. In line 214, the phrase "...scores equal or greater than twelve..." should probably be changed to "...scores equal to or greater than twelve..."
  5. In line 291, "Under of doxorubicin treatment...." should probably be changed to "Under doxorubicin treatment...." 
  6. In line 424, "resist" should probably be changed to "resistant"

Author Response

July 9, 2021

Editor

Cells

RE: cells-1286500

Dear Editor:

Thank you for your letter on July 2, 2021, inviting us to submit a revised version of our manuscript ID# cells-1286500 “CK2α/CSNK2A1 induces resistance to doxorubicin through SIRT6 mediated activation of the DNA damage repair pathway” by Hussein et al.

We appreciate the careful review of the manuscript by the reviewers and in response, we have introduced all the suggested changes and revisions. We thank the reviewers and the editor and feel that the suggested revisions have significantly strengthened the paper. Please find a point-by-point response to each of the concerns posed by the four reviewers below.

Revisions introduced into the text are in red.

Thank you for your kind consideration.

Sincerely yours,

Kyu Yun Jang, MD, PhD

Address: Department of Pathology, Jeonbuk National University Medical School, 567 Baekje-daero, Dukjin-gu, Jeonju, 54896, Republic of Korea

Tel: +82-63-270-3071, Fax: +82-63-270-3135, E-mail: [email protected]

Jung Ryul Kim, MD, PhD

Address: Department of Orthopedic Surgery, Jeonbuk National University Medical School, 567 Baekje-daero, Dukjin-gu, Jeonju, 54896, Republic of Korea

Tel: +82-63-250-1767, Fax: +82-63-271-6538, E-mail: [email protected]

See-Hyoung Park, PhD, Associate Professor

Address: Department of Bio and Chemical Engineering, Hongik University, Sejong, 30016, Republic of Korea

Tel: +82-44-860-2126, E-mail: [email protected]

Response to reviewer 1

We thank the reviewer for these insightful comments.

Reviewer reports:

The manuscript by Hussein et al. reports findings on the role of the CSNK2A1-SIRT6 axis in the chemotherapy response of osteosarcoma (OS) cells. The constitutively active Ser/Thr protein kinase CSNK2A1 phosphorylates hundreds of substrates and controls several signaling pathways. It has been implicated, among many other conditions, in cancer, where it regulates practically all malignant hallmarks. CSNK2A1-mediated SIRT6 phosphorylation has also been reported in the progression of other cancers. The novelty of the current paper is that it systematically evaluates the role of the CSNK2A1-SIRT6 axis in OS and unequivocally demonstrates that this pathway determines the outcome of doxorubicine-treatment of OS cells. The findings suggest that CSNK2A1 might be used as a prognostic indicator in osteosarcoma patients, especially in patients who received postoperative chemotherapy. Furthermore, the authors have shown that CSNK2A1 induces resistance to DOX through SIRT6 phosphorylation-mediated activation of the DNA damage repair pathway in vitro and in vivo.

The paper is well written and easy to follow. The figures were carefully designed although the figure texts are often difficult to read due to very small font sizes (e.g. on figures 3, 5, 6, 7). The conclusions are supported by the data.

We thank the reviewer for this comment. In response to the comment of the reviewer, we have increased the font size in the figures.

Comments

  1. The authors chose DOX as a chemotherapeutic drug to study in cell culture experiments. It is recommended to justify this choice in the paper. Was DOX a part of the chemotherapy given to patients enrolled in the study? What else was included in the complex chemotherapy of the patients?

We thank the reviewer for this comment. In our patients, standard chemotherapy was triple combined chemotherapy with doxorubicin, high-dose methotrexate, and cisplatin. In response to the comment of the reviewer, we have included this information in the revised manuscript. Below are the revised sentences in the manuscript.

Materials and Methods section

Twenty-six patients received adjuvant chemotherapy (doxorubicin, high-dose methotrexate, and cisplatin).

  1. All Western blots should be evaluated by densitometry followed by statistical analysis. In the current version of the MS, only representative WB images are presented (Fig 4A and 5A).

We are very grateful to the reviewer for this comment. In response to the reviewer’s comment, we performed quantitative analysis for the bands of the western blot in Figures 4 and 5. Below are the revised figures.

Revise Figure 4

Figure 4. CSNK2A1 involvement in the apoptotic effects of doxorubicin in osteosarcoma cells with western blot analysis. U2OS and KHOS/NP osteosarcoma cells were transfected with control shRNA, shRNA for CSNK2A1, empty vector, or wild-type CSNK2A1 and treated with 0.2 μM doxorubicin. The expression patterns of CSNK2A1, PARP1, cleaved PARP1, cleaved caspase 3, BCL2, BAX, and GAPDH were determined via western blot. The density of the western blot bands was measured in triplicate by using ImageJ software. *; p < 0.05, **; p < 0.001.

Revised Figure 6

Figure 6. CSNK2A1 involvement in doxorubicin resistance through SIRT6 mediated activation of the DNA damage repair pathway with western blot analysis. U2OS and KHOS/NP cells were transfected with control shRNA, shRNA for CSNK2A1, empty vector, or WT-CSNK2A1 and treated with DMSO or 0.2 μM doxorubicin. Western blots for CSNK2A1, SIRT6, phosphorylated SIRT6 (pSIRT6, Ser338), phosphorylated ATM (pATM), phosphorylated Chk2 (pChk2), phosphorylated H2AX (γH2AX, ser138), and GAPDH were performed, and the density of the bands was measured in triplicate by using ImageJ software. **; p < 0.001.

  1. Since the MS contains very large amounts of data, the reviewer is hesitant to suggest further experiments. However, measuring one more parameter [i.e. poly(ADP-ribose) polymer] would add important mechanistic information to the role of the CSNK2A1-SIRT6 axis in the DNA damage response. Showing PAR polymer levels after CSNK2A1 overexpression/silencing would be useful especially in light of the fact that SIRT6 is known to activate PARP1.

We thank the reviewer for this comment and agree with the reviewer. However, an anti-PAR antibody was not available to us in time. Therefore, in response to the reviewer’s comment and instead for PAR, we have performed western blots for PARP1 to show total levels of PARP1. According to these new results, we have revised the related sections of the revised manuscript.

  1. In line 214, the phrase "...scores equal or greater than twelve..." should probably be changed to "...scores equal to or greater than twelve..."

We are very thankful to the reviewer for this comment. In response to the reviewer’s comment, we have revised our manuscript.

  1. In line 291, "Under of doxorubicin treatment...." should probably be changed to "Under doxorubicin treatment...." 

We are very thankful to the reviewer for this comment. In response to the reviewer’s comment, we have revised our manuscript.

  1. In line 424, "resist" should probably be changed to "resistant"

We are very thankful to the reviewer for this comment. In response to the reviewer’s comment, we have revised our manuscript.

Reviewer 2 Report

The manuscript by Khamis Hussein reports the role of CSNK2A1 and SIRT6 during the chemorresistance to doxorubicin treatment in osteosarcoma patients and human cells lines. They confirm that CSNK2A1 expression correlated with poor survival in osteosarcoma tumour samples. CSNK2A1 overexpression Induced resistance to doxorrubicine whereas CSNK2A1 loss of function experiments, throughout its knockdown or inhibition sensitized cells to chemotherapy. Moreover CSNK2A1-mediated chemorresistance was associated to its ability to induce ser338-SIRT6 phosphorylation, suggesting that CSNK2A1 inhibition might be a novel therapeutical choice for osteosarcoma patients. Overall, the study is well conducted and the conclusion are neat. However, there are some concerns about the number of replicates and some experimental experiment that needs to be performed to recommend publication at Cells journal

For most of the experiments carried out by the authors, the number of replicates each experiment have been carried out independently is missing. Experiments have to be carried out at least n=3 with the proper statistical analysis.

Figure 1: The authors show that the CSNK2A1 immunohistological staining of osteosarcoma samples with a cut-off of 12, CSNK2A1 expression inversely correlated with OS and RFS (figure 1C). The authors stressed that CSNK2A1 mediates doxorubicin resistance through phosphorylation of SIRT6 at ser338. Have the authors determined ser338-SIRT6 in the TMA and examined if the OS and RFS correlated with ser338-SIRT6 immunostaining?

Figure 3a, 3b, 3c,5c: Proliferation assay units are shown in Absorbance 560nm. The data should be represented relative to 100.

Related: Tin figure 3a, the differences in proliferation of EV versus CSNK2A1 cell lines upon dox treatment described in figure 3a are negligible and do not increase over time. Indeed, under these conditions cells did not proliferate over time at all (compare 24 vs 72 hours. If so, there were less viable cells at 72h compared to 24h. Has the authors studied cell cycle populations in G1,S and G2 of EV and CSNK2A1 cell lines? Because Dox preferentially kills cells S phase, the effects described could be ascribed to differences in cell cycle populations. Alternatively, lower dose of Dox could be used to allow a marginal cell proliferation.

Figure 3b and 3d should describe the Dox treatment length. Were cells pulsed with Dox for 1hours, for 24hours or for the whole time?

In figure 4b, in agreement with the author´s hipothesis, the authors shows that Dox treatment increased AnnexinV-mediated cell apoptotis of shCSNK2A1 compared to shControl. However the authors did not show the apoptosis analysis of EV vs CSNK2A1 overexpressing cells. To be consistent, this analysis should be carried out to strengthen their hypothesis.

Figure 7. Emodin treatment efficiently decrease cell growth at the dose used. However, I don´t know how specific emodin-mediated growth inhibition for osteosarcoma cells compared to other non-osteosarcoma cells. Does emodin (0.5mM) inhibit cell growth of non-osteosarcoma cells? Or does that emodin concentration halt cell growth of any cell line? Related to this, why emodin was not included in the xenograft experiment? This could have been quite informative about the potential use of emodin in osteosarcoma patient treatment.

Minor comments:

Lane 268: “overexpression of CSNK2A1 has not affected…” should be change to “overexpression of CSNK2A1 did not affect…”.

Lane 491: “Archive” should be change to “achive”

Author Response

July 9, 2021

Editor

Cells

RE: cells-1286500

Dear Editor:

Thank you for your letter on July 2, 2021, inviting us to submit a revised version of our manuscript ID# cells-1286500 “CK2α/CSNK2A1 induces resistance to doxorubicin through SIRT6 mediated activation of the DNA damage repair pathway” by Hussein et al.

We appreciate the careful review of the manuscript by the reviewers and in response, we have introduced all the suggested changes and revisions. We thank the reviewers and the editor and feel that the suggested revisions have significantly strengthened the paper. Please find a point-by-point response to each of the concerns posed by the four reviewers below.

Revisions introduced into the text are in red.

Thank you for your kind consideration.

Sincerely yours,

Kyu Yun Jang, MD, PhD

Address: Department of Pathology, Jeonbuk National University Medical School, 567 Baekje-daero, Dukjin-gu, Jeonju, 54896, Republic of Korea

Tel: +82-63-270-3071, Fax: +82-63-270-3135, E-mail: [email protected]

Jung Ryul Kim, MD, PhD

Address: Department of Orthopedic Surgery, Jeonbuk National University Medical School, 567 Baekje-daero, Dukjin-gu, Jeonju, 54896, Republic of Korea

Tel: +82-63-250-1767, Fax: +82-63-271-6538, E-mail: [email protected]

See-Hyoung Park, PhD, Associate Professor

Address: Department of Bio and Chemical Engineering, Hongik University, Sejong, 30016, Republic of Korea

Tel: +82-44-860-2126, E-mail: [email protected]

Response to reviewer 2

We thank the reviewer for their insightful comments.

Comments to Author:

The manuscript by Khamis Hussein reports the role of CSNK2A1 and SIRT6 during the chemoresistance to doxorubicin treatment in osteosarcoma patients and human cells lines. They confirm that CSNK2A1 expression correlated with poor survival in osteosarcoma tumour samples. CSNK2A1 overexpression Induced resistance to doxorubicine whereas CSNK2A1 loss of function experiments, throughout its knockdown or inhibition sensitized cells to chemotherapy. Moreover CSNK2A1-mediated chemoresistance was associated to its ability to induce ser338-SIRT6 phosphorylation, suggesting that CSNK2A1 inhibition might be a novel therapeutical choice for osteosarcoma patients. Overall, the study is well conducted and the conclusion are neat. However, there are some concerns about the number of replicates and some experimental experiment that needs to be performed to recommend publication at Cells journal.

We thank the reviewer for this comment.

For most of the experiments carried out by the authors, the number of replicates each experiment have been carried out independently is missing. Experiments have to be carried out at least n=3 with the proper statistical analysis.

 We thank the reviewer for this comment. All experiments were performed three times, and representative data are presented with the mean ± standard deviation. In response to the comment of the reviewer, we have revised the manuscript to make this clear. Below are the revised sentences in the manuscript.

Materials and Methods section

All experiments were performed in triplicate and performed three times, with representative data presented.

Figure 1: The authors show that the CSNK2A1 immunohistological staining of osteosarcoma samples with a cut-off of 12, CSNK2A1 expression inversely correlated with OS and RFS (figure 1C). The authors stressed that CSNK2A1 mediates doxorubicin resistance through phosphorylation of SIRT6 at ser338. Have the authors determined ser338-SIRT6 in the TMA and examined if the OS and RFS correlated with ser338-SIRT6 immunostaining?

We thank the reviewer for this comment. In response to the reviewer’s comment, we performed additional immunohistochemical staining for pSIRT6 (Ser338) in the TMA and analyzed the results. According to these new results, we have revised the related sections of the revised manuscript. Below are the major sections of the manuscript revised according to these new results.

Results section

To evaluate the clinicopathological significance of the expression of CSNK2A1 and pSIRT6 in human osteosarcomas, we performed immunohistochemical staining for CSNK2A1 and pSIRT6. Representative images of the immunohistochemical expression pattern of CSNK2A1 and pSIRT6 are presented in Figure 1a. The positivity of the immunohistochemical expression of CSNK2A1 and pSIRT6 were determined with receiver operating characteristic curve analysis (Figure 1b). The cut-off point was determined at the point with the highest area under the curve to predict the death of osteosarcoma patients (Figure 1b). The cut-off points for both CSNK2A1 and pSIRT6 were twelve, and the cases with immunohistochemical staining scores equal to or greater than twelve were considered positive for CSNK2A1 immunostaining (Figure 1b). With these cut-off points, CSNK2A1-positivity was significantly associated with sex (p = 0.046), higher tumor stage (p = 0.006), higher T category (p = 0.028), higher M category (p = 0.047), latent distant metastasis (p = 0.025), and pSIRT6-positivity (p < 0.001) (Table 1). Positivity for pSIRT6 was significantly associated with sex (p = 0.014), tumor size (p = 0.031), higher T category (p = 0.011), and latent distant metastasis (p = 0.001) (Table 1).

In univariate survival analysis, age (OS; p = 0.039, RFS; p = 0.018), tumor size (OS; p = 0.015, RFS; p = 0.019), tumor stage (OS; p = 0.015, RFS; p = 0.005), T category (OS; overall p = 0.059, RFS; overall p = 0.020), M category (OS; p = 0.007, RFS; p = 0.018), pSIRT6 expression (OS; p = 0.004, RFS; p = 0.001), and CSNK2A1 expression (OS; p = 0.002, RFS; p < 0.001) were significantly associated with OS or RFS (Table 2). Positivity of pSIRT6 expression predicted a 6.269-fold (95% confidence interval [95% CI]; 1.807-21.750) greater risk of death and a 7.783-fold (95% CI; 2.242-27.019) greater risk of relapse or death of osteosarcoma patients (Table 2). CSNK2A1-positivity predicted a 10.081-fold (95% CI; 2.307-44.054) greater risk of death and a 12.179-fold (95% CI; 2.777-53.407) greater risk of relapse or death of osteosarcoma patients (Table 2). The Kaplan-Meier survival curves for OS and RFS of CSNK2A1 and pSIRT6 expression are presented in Figure 1c.  

In the univariate analysis of 26 osteosarcoma patients who received adjuvant chemotherapy, the expression of CSNK2A1 (OS; p = 0.008, RFS; p = 0.003) and pSIRT6 (OS; p = 0.015, RFS; p = 0.008) were significantly associated with OS and RFS (Table 4) (Figure 2). In the multivariate analysis performed with the inclusion of age, tumor size, stage, T category, N category, M category, pSIRT6 expression, and CSNK2A1 expression, N category was an independent indicator of OS (p = 0.029), pSIRT6 expression was an independent indicator of OS (p = 0.011), and CSNK2A1 expression was an independent indicator of RFS (p = 0.003) (Table 5). Positivity of pSIRT6 expression predicted an 18.649-fold (95% CI; 1.949-178.412) greater risk in OS analysis, and CSNK2A1 expression predicted a 10.374-fold (95% CI; 2.244-47.968) greater risk in RFS analysis in osteosarcoma patients who received adjuvant chemotherapy (Table 4).

Figure 1

Figure 1. Immunohistochemical expression of CSNK2A1 and pSIRT6 in human osteosarcoma and statistical and survival analysis in osteosarcoma. (a) Negative and positive examples of immunohistochemical expression of CSNK2A1 and pSIRT6 in human osteosarcoma tissue. (b) The cut-off points of CSNK2A1 and pSIRT6 expression were determined using receiver operating characteristic curve analysis. The cut-off points of the immunohistochemical staining score for CSNK2A1 and pSIRT6 were twelve for both markers, and was determined at the point with the highest area under the curve (AUC). The red arrow (CSNK2A1) and blue arrow (pSIRT6) indicate the cut-off points. (c) Kaplan-Meier survival curves for overall survival and relapse-free survival according to the expression of CSNK2A1 and pSIRT6 in 37 osteosarcoma patients.

Figure 2

Figure 2. Kaplan-Meier survival curves for overall survival and relapse-free survival according to expression of CSNK2A1 and pSIRT6 in 26 osteosarcoma patients who received adjuvant chemotherapy.

Table 1

Table 1. The clinicopathological characteristics of 37 osteosarcomas and their association with the immunohistochemical expression of CSNK2A1.

Characteristics

No.

CSNK2A1

pSIRT6

Positive

p

Positive

p

Age, years

< 30

24

11 (46%)

0.068

12 (50%)

0.501

≥ 30

13

10 (77%)

8 (62%)

Sex

Male

25

17 (68%)

0.046

17 (68%)

0.014

Female

12

4 (33%)

3 (25%)

Tumor size

≤ 8 cm

19

8 (42%)

0.065

7 (37%)

0.031

> 8 cm

18

13 (72%)

13 (72%)

Stage

I & II

26

11 (42%)

0.006

12 (46%)

0.138

III & IV

11

10 (91%)

8 (73%)

T category

1

17

6 (35%)

0.028

5 (29%)

0.011

2

16

11 (69%)

11 (69%)

3 and 4

4

4 (100%)

4 (100%)

N category

N0

34

19 (56%)

0.718

19 (56%)

0.452

N1

3

2 (67%)

1 (33%)

M category

M0

29

14 (48%)

0.047

15 (52%)

0.588

M1

8

7 (88%)

5 (63%)

Latent distant metastasis

Absence

28

13 (46%)

0.025

11 (39%)

0.001

Presence

9

8 (89%)

9 (100%)

pSIRT6

Negative

17

3 (18%)

< 0.001

Positive

20

18 990%)

Table 4. Univariate and multivariate Cox proportional hazards regression analysis of the survival of 26 osteosarcoma patients who received adjuvant chemotherapy.

Characteristics

No.

OS

RFS

HR (95% CI)

p

HR (95% CI)

p

Univariate analysis

CK2α, positive (vs. negative)

13/26

7.741 (1.693-35.408)

0.008

10.374 (2.224-47.968)

0.003

pSIRT6, positive (vs. negative)

15/26

12.682 (1.639-98.119)

0.015

15.709 (2.035-121.271)

0.008

Multivariate analysis

N category, N1 (vs. N0)

29.727 (1.425-620.194

0.029

pSIRT6, positive (vs. negative)

18.649 (1.949-178.412)

0.011

CK2α, positive (vs. negative)

10.374 (2.244-47.968)

0.003

Figure 3a, 3b, 3c, 5c: Proliferation assay units are shown in Absorbance 560nm. The data should be represented relative to 100.

We thank the reviewer for this comment. In response to the reviewer’s comment, we have presented data as “% of control” and revised the figures.

Related in figure 3a, the differences in proliferation of EV versus CSNK2A1 cell lines upon dox treatment described in figure 3a are negligible and do not increase over time. Indeed, under these conditions cells did not proliferate over time at all (compare 24 vs 72 hours. If so, there were less viable cells at 72h compared to 24h. Has the authors studied cell cycle populations in G1, S and G2 of EV and CSNK2A1 cell lines? Because Dox preferentially kills cells S phase, the effects described could be ascribed to differences in cell cycle populations. Alternatively, lower dose of Dox could be used to allow a marginal cell proliferation.

We thank the reviewer for this comment. In this study, we used doxorubicin because doxorubicin is the component of conventional chemotherapeutic agent in osteosarcoma patients and it induces DNA damage. The main proposed mechanisms of anticancer pharmacodynamics of doxorubicin have been its intercalation into DNA and disruption of topoisomerase-II-mediated DNA repair and it stimulates generation of free radicals and their damage to cellular membranes, DNA and proteins. Therefore, based on the anti-cancer mechanisms of doxorubicin, we evaluated the effect of CSNK2A1 expression and SIRT6 phosphorylation on ser338. To support the hypothesis of this study we needed to evaluate death of osteosarcoma cells and all data should be consistent with diverse experimental methodology (proliferation, western blot, and flow cytometry apoptosis analysis). Therefore, based on our preliminary analysis we determined that we should use 0.2 μM doxorubicin. In the data of Figure 3a, the decrease in cell number with treatment of doxorubicin is associated with the cytotoxic effect of doxorubicin and the cytotoxic effectiveness is also related to the treatment dose of doxorubicin. Use of 0.2 μM doxorubicin might be relatively high; however, to clearly present the mechanism of CSNK2A1-mediated resistance to doxorubicin-mediated cytotoxicity, we used 0.2 μM doxorubicin, which relatively higher than the dose of marginal effect on proliferation.

Figure 3b and 3d should describe the Dox treatment length. Were cells pulsed with Dox for 1 hours, for 24 hours or for the whole time?

The cells were grown for ten days. In the colony forming assay, the cells were grown for ten days. In response to the comment of the reviewer, we have revised the figure legends. In addition, in in vitro experiments, doxorubicin was added initially when we seeded cells and maintained in the culture media.

Figure legends

Figure 3. ... (b) For the colony-forming assay, U2OS (1×103) and KHOS/NP (1×103) cells transfected with empty vector or wild-type CSNK2A1 were treated with DMSO or 0.2 μM doxorubicin in 24-well culture plates. The cells were grown for ten days. Clono-Counter software was used in the quantification of the number of colonies. (d) For the colony-forming assay, U2OS (1×103) and KHOS/NP (1×103) cells transfected with control shRNA or shRNA for CSNK2A1 were treated with DMSO or 0.2 μM doxorubicin in 24-well culture plates. The cells were grown for ten days.

In figure 4b, in agreement with the author´s hipothesis, the authors shows that Dox treatment increased AnnexinV-mediated cell apoptotis of shCSNK2A1 compared to shControl. However the authors did not show the apoptosis analysis of EV vs CSNK2A1 overexpressing cells. To be consistent, this analysis should be carried out to strengthen their hypothesis.

We thank the reviewer for this comment. In response to the reviewer’s comment, we have presented data for apoptosis analysis with the cells overexpressing CSNK2A1 and revised Figure 4b. Below is the revised Figure 5.

Figure 5

Figure 5. CSNK2A1 involvement in the apoptotic effects of doxorubicin in osteosarcoma cells with flow cytometry analysis. (a) U2OS and KHOS/NP osteosarcoma cells were transfected with control shRNA or shRNA for CSNK2A1 and treated with 0.2 μM doxorubicin for 24 hours. (b) U2OS and KHOS/NP osteosarcoma cells were transfected with empty vector or wild-type CSNK2A1 and treated with 0.2 μM doxorubicin for 24 hours. Apoptosis of cells was evaluated via flow-cytometry analysis with staining with propidium iodide and for annexin V. **; p < 0.001.

Figure 7. Emodin treatment efficiently decrease cell growth at the dose used. However, I don´t know how specific emodin-mediated growth inhibition for osteosarcoma cells compared to other non-osteosarcoma cells. Does emodin (0.5mM) inhibit cell growth of non-osteosarcoma cells? Or does that emodin concentration halt cell growth of any cell line? Related to this, why emodin was not included in the xenograft experiment? This could have been quite informative about the potential use of emodin in osteosarcoma patient treatment.

We thank the reviewer for this comment. In this study, we focused on the mechanism of CSNK2A1 mediated phosphorylation of SIRT6 on Ser338 in chemoresistance. Therefore, we performed xenograft experiments after inducing overexpression of WT CSNK2A1, WT SIRT6, and SIRT6-S338A. Thereafter, based on this mechanism, we evaluated the anticancer effect of CSNK2 inhibitor in vitro. In addition, there are several CSNK2 inhibitors and they have anticancer effects on various types of cancer cells. In our previous reports on the role of CSNK2A1 for the phosphorylation of SIRT6, as we discussed in the discussion section, we have evaluated emodin and CX4945 in breast cancer cells. In addition, we agree with the reviewer that in vivo evaluation of the synergistic effect of emodin with doxorubicin might be very helpful in osteosarcoma patients. Therefore, further evaluation is needed to clearly show the effectiveness of blocking the CSNK2A1-SIRT6 pathway in chemoresistance in osteosarcomas by using various inhibitors of the CSNK2A1-SIRT6 pathway. Concerning this point, we have described this in the discussion section.

However, this study has the limitations that we have used just one CSNK2A1 inhibitor and did not evaluate the effectiveness of emodin in vivo. Therefore, additional studies evaluating the effectiveness of various inhibitors of CSNK2A1 in osteosarcoma are needed.

Minor comments:

Lane 268: “overexpression of CSNK2A1 has not affected…” should be change to “overexpression of CSNK2A1 did not affect…”.

We are very thankful to the reviewer for this comment. In response to the reviewer’s comment, we have revised our manuscript.

Lane 491: “Archive” should be change to “achieve”

We are very thankful to the reviewer for this comment. In response to the reviewer’s comment, we have revised our manuscript.

Round 2

Reviewer 2 Report

The author have now provided compelling evidences for the role of CSNK2A1 and pSIRT6 on the resistance to doxorubicin, and their manuscript has improved in quality. I therefore support its publication.